# Brief communication: Radar images for monitoring informal urban settlements in vulnerable zones in Lima, Peru

Luis Moya[1,3], Fernando Garcia[2], Carlos Gonzales[1], Miguel Diaz[1], Carlos Zavala[1], Miguel Estrada[1], Fumio Yamazaki[4], Shunichi Koshimura[3], Erick Mas[3], and Bruno Adriano[5]

[1]Japan-Peru Center for Earthquake Engineering Research and Disaster Mitigation, National University of Engineering, Tupac Amaru Avenue 1150, Lima 25, Peru
[2]Graduate School of Civil Engineering, National University of Engineering, Av. Túpac Amaru 280, Lima 25, Peru
[3]International Research Institute of Disaster Science, Tohoku University, Aoba 468-1, Aramaki, Aoba-ku, Sendai 980-8572, Japan
[4]National Research Institute for Earth Science and Disaster Resilience, Tsukuba, Ibaraki 305-0006, Japan
[5]Geoinformatics Unit, RIKEN Center for Advanced Intelligence Project (AIP)

**Correspondence:** Luis Moya (lmoyah@uni.pe)

**Abstract.** Lima city, Peru's capital, has about 9.6 million inhabitants and keeps attracting more residents searching for a better life. Many citizens, without access to housing subsidies, live in informal housing and shack settlements. A typical social phenomenon in Lima is the sudden illegal occupation of areas for urban settlements. When such areas are unsafe against natural hazards, it is important to relocate such a population to avoid significant future losses. In this communication, we present an application of Sentinel-1 SAR images to map the extension of a recent occupation of an area with unfavorable soil conditions against earthquakes.

## 1 Introduction

Urban sprawl in Latin America has been influenced by the people migration from rural areas to the cities, which have produced high-density urban areas (United Nations, 2019). Informal settlement refers to the organization of people in search of housing who occupy unused land and perform collective actions to self-resolve their urban and social organization issues (Kapstein and Aranda, 2014). The urban growth in the capital city of Peru during the 20th century was mainly driven by informal urban expansion, which was motivated by the government policy of allowing people with low socioeconomic status to occupy unused land. Currently, the oldest informal settlements in Lima have obtained basic services, such as electricity and water. However, they are still vulnerable in terms of crime and security, accessibility, and natural hazards. In recent years, informal settlements have kept increasing countrywide. According to the Ministry of Housing, Construction and Sanitation (2021), 93% of the urban growth in Peru between 2001 and 2018 consisted of informal settlements. Frequently, informal settlements occupy unsafe zones against natural hazards. For instance, Müller et al. (2020) showed that slum residents are more likely to settle in areas exposed to landslides than formal residents. Furthermore, during the Niño Costero phenomenon in 2017, 63800 houses were destroyed due to river overflows. An economic loss of about $3124 million was estimated.

Remote sensing data have been used to extract information from urban and rural areas, such as land cover classification (Geiß et al., 2020), urban growth (Shi et al., 2019), and detection of damaged buildings (Moya et al., 2021). Regarding informal settlements, a comprehensive study of its spatial morphology can be found in Taubenböck et al. (2018). A further study to consider the temporal effects is reported in Kraff et al. (2020). However, remote sensing studies to identify spontaneous informal settlements consisting of makeshift shelters are scarce (Kuffer et al., 2016). The relevance of this task is the geolocation of

such settlements to perform prompt prevention actions, such a relocation when the settlements are located in vulnerable areas.

This study shows how a simple, yet effective, time series analysis of radar images from the Sentinel-1 constellation can map spontaneous informal settlements. We report a recent case in Lima, where a group of people settled in two hazardous unused land spaces. The rest of the manuscript is structured as follows. The following section reports a summary of the events that occurred in the two areas. Section 3 shows the time series analysis of the radar images. Finally, our conclusions are drawn in

Section 4.

## 2   The urban settlement of Lima

Lima metropolitan face very frequently critical problems regarding the illegal occupation of unused land and land trafficking. The most recent occurred in April 2021, when two large areas in the outskirts of Lima were informally occupied by citizens (Figure 1). The people who participated declared to the local news that they were affected by the pandemic and cannot afford their rents any longer. The Morro Solar, located in the Chorrillos district, was first occupied on April 5, 2021. Then, the Lomo

de Corvina sector, located in Villa El Salvador District, was occupied on April 12, 2021. According to the local news, about 5000 and 3000 people occupied the Morro Solar and the Lomo de Corvina, respectively. During this sudden occupation, lots were delimited with chalk, and makeshift shelters were built with wooden sticks and plastic sheets (See Figures 1e and 1f). It is worth mentioning that the government declared the Morro Solar as an intangible area in 1977 and a historical monument in 1986. The police, in coordination with the local authorities, removed the inhabitants that participated in the invasions in April

14 and April 28 in Morro Solar and Lomo de Corvina, respectively, and the makeshift shelters were removed several days later.

The main concern this study addresses is that the occupied areas are unsafe against natural hazards. Luque and Núñes (2011) reported a geological evaluation of Inty Llacta, an informal settlement located close to the recently invaded area in Morro Solar. The referred report identified rockfalls and slope failure hazards, representing a severe danger to the inhabitants and

buildings. Likewise, Medina et al. (2013) reported a technical evaluation of twenty-nine informal settlements located in the Lomo de Corvina sector and pointed out potential slope failure may occur at the referred areas. It was also mentioned that the west part of the Lomo de Corvina, where the recent invasion was located, is the most critical area and is not suitable for urban development.

Regarding seismic hazards, seismic microzonation studies have been performed in Metropolitan Lima, which included the

implementation of field tests in the city's urban areas. Within the framework of these studies, the underlying soil deposits that conform to the capital city were classified into different zones depending on their mechanic and dynamic behavior. Thus, Zone I corresponds to the stiffest soil of the city and Zones IV and V to the most inadequate regions to be populated due to their

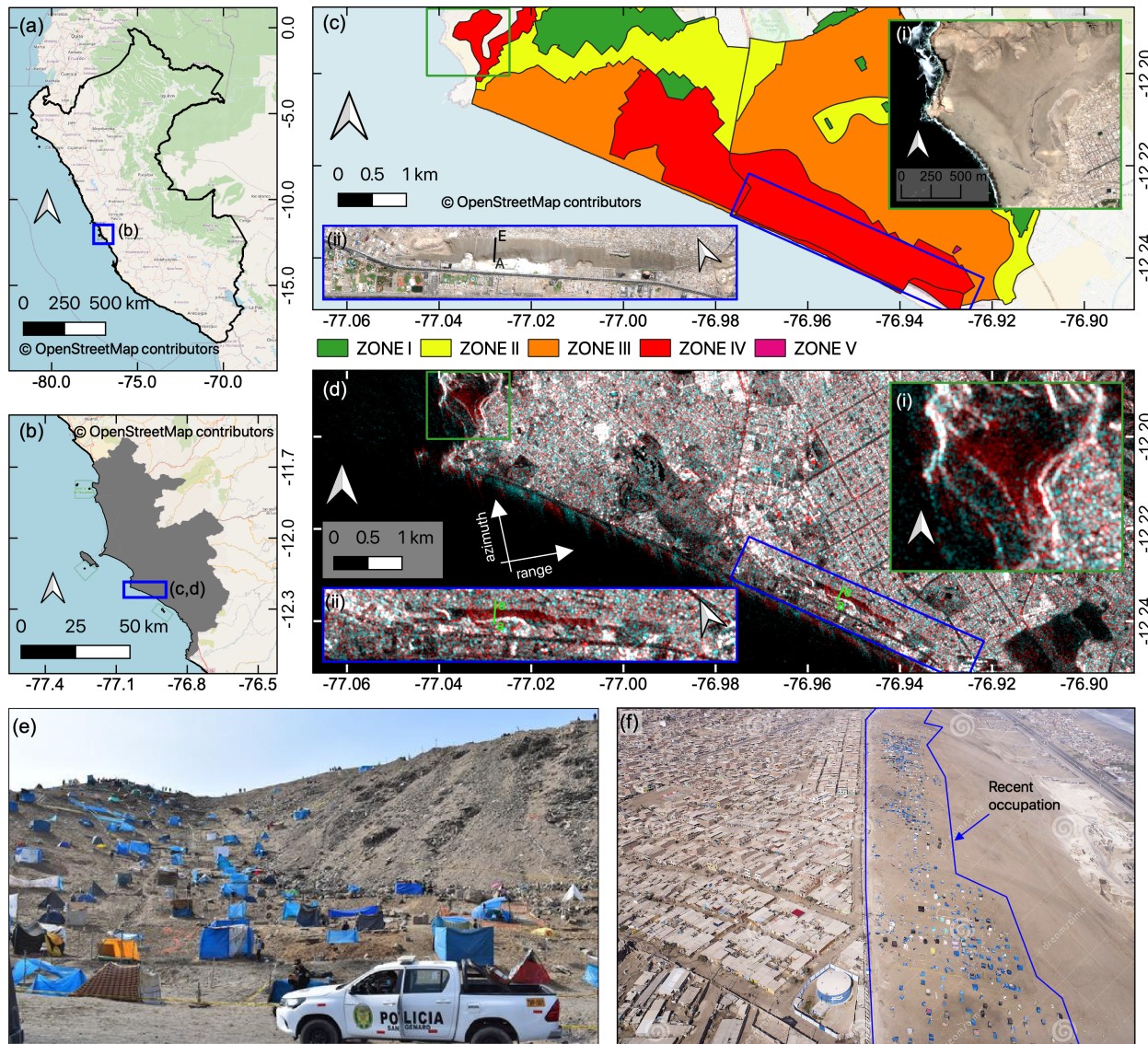

**Figure 1.** Location of the study area. (a) Location of Metropolitan Lima (blue rectangle) within Peru. (b) Location of the study area (blue rectangle) within Metropolitan Lima, which is partly located in the districts of Chorrillos and Villa el Salvador. (c) Seismic microzonation of the study area. The green rectangle denotes the location of inset (i), the Morro Solar. The blue rectangle denotes the location of the inset (ii), the Lomo de Corvina. Basemap of insets: MAXAR imagery. (d) Color composite of SAR backscattering intensity images. Red band: image recorded on April 14, 2021; Green and blue band: image recorded on December 03, 2020. (e) Photograph of the squatter settlement in Morro Solar recorded by Gestion (2021). (f) Photograph of the squatter settlement in Lomo de Corvina (modified from Inga (2021)).

particular bad soil conditions. In the case of Chorrillos district, it was divided into four zones, and the areas in Morro Solar are categorized as Zone IV mainly because of the steep slopes that might fail under the effect of strong motions (CISMID, 2010).

Furthermore, Lomo de Corvina is also classified as Zone IV due to the existence of deep eolian sand deposits and the largest fundamental vibration period found in Lima city, which is slightly larger than 1 s (CISMID, 2011). Recent studies have shown that Lomo de Corvina might evidence important values of amplification factors due to the generation and interference of surface waves along the slope (Gonzales et al., 2019). It is worth mentioning that the collapse of the light makeshifts built during the recent invasions may not have represented an effective danger condition to the inhabitants. However, non-engineering masonry

houses could have been constructed in the short term if the inhabitants were not removed. Note, however, that the sole action of removing shelters will exacerbate the need for residence.

## 3    Informal settlements from earth observation technologies

This section shows how the informal settlements were observed from satellite imagery. Unfortunately, the areas of interest were cloud-covered in most of the days after the invasion. From the beginning of April 2021 until May 5, only one product from

the Sentinel-2 constellation recorded on April 20 was cloud-free. On the other hand, synthetic aperture radar (SAR) images are designed to pass through the clouds; thus, we focus on SAR images from the Sentinel-1 constellation. A total of fourteen Sentinel-1 SAR images, recorded from December 3, 2020, to May 8, 2021, were used for the analysis. The images were taken by the VV polarization from the ascending path, and the incident angle at the two study areas are about $38°$. Figure 1d shows a color composite of backscattered intensity recorded on different dates. The red band denotes an image recorded on April 14,

2021, and the green and blue band denote the image recorded on December 03, 2020. Note from inset (ii) in Figures 1c and 1d that the Lomo de Corvina area in the SAR image looks smaller than that from the optical image. Such geometric distortions are because of the oblique observation geometry of SAR images. Figure 2a depicts a scheme of the elevation profile corresponding to the line $\overline{AE}$ depicted in the inset (ii) of Figure 1. The profile is simplified by four segments, $\overline{AB}$, $\overline{BC}$, $\overline{CD}$, and $\overline{DE}$. In order to illustrate the geometric distortions in the SAR images, it is assumed that the range direction is parallel to the vertical

plane that contains the profile. The slant range denotes the direction from which the microwave energy travels from the satellite to the ground. The microwave images are presented in ground-range format. Note that because points A and C share about the same distance to the satellite, both segments $\overline{AB}$ and $\overline{CB}$ occupy the same geographic position in the ground-range ($\overline{ab}$ and $\overline{cb}$) of the SAR image. In the optical image, the profiling scheme is located over the line $\overline{AE}$ (Figure 1c); on the other hand, the profile is located along the segment $\overline{ae}$ in the SAR image, which is significantly smaller. However, the segment that contains

the informal settlement in the optical image, $\overline{DE}$, is only slightly larger than the length of $\overline{de}$. Note also that there is a shift between $\overline{DE}$ and $\overline{de}$.

Regarding the color composite SAR image shown in Figure 1c, red tones are observed in the new informal settlements, which express an increment of the backscattering in time. Such increment is originated from the double bounce scattering mechanism produced by the shelters. Figures 2b and 2c depict a scheme of the backscattering mechanism under two scenarios.

When the radar pulses emitted by the satellite reach a flat area, only a small fraction of the energy is backscattered to the sensor

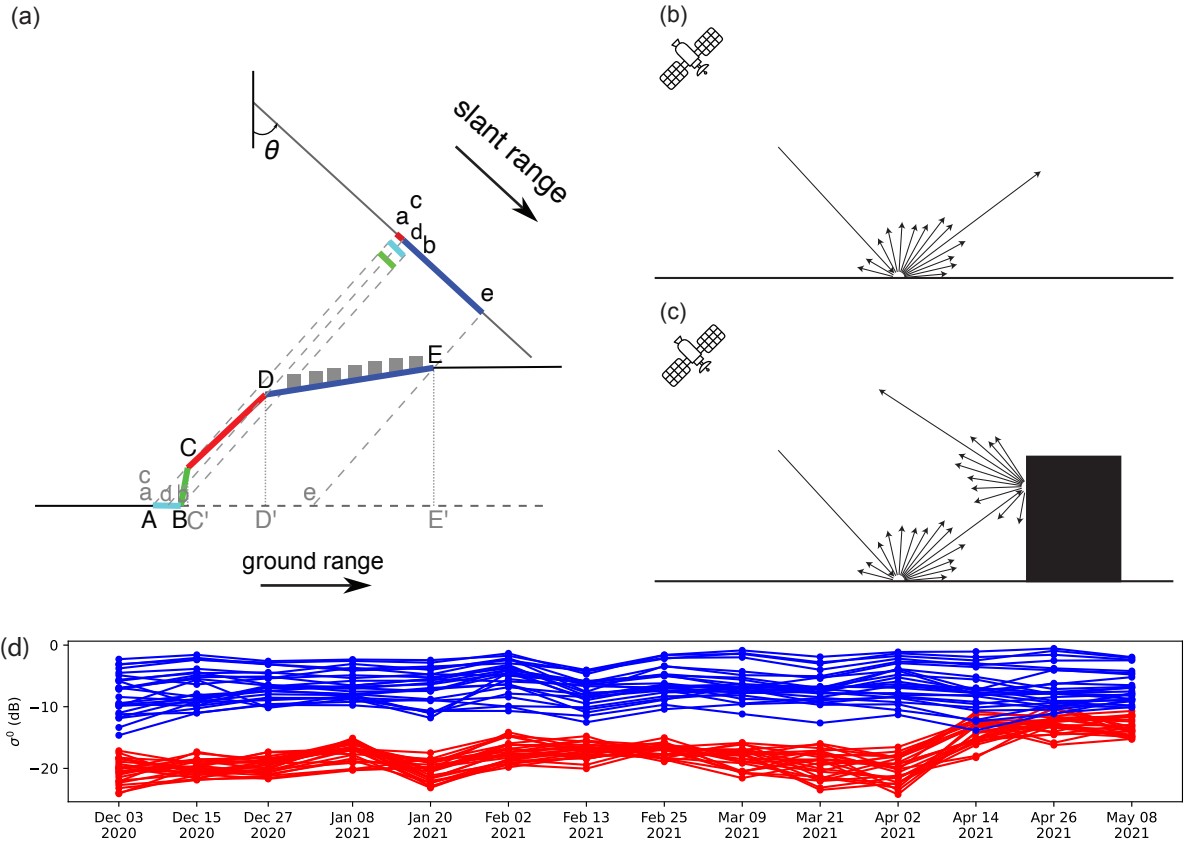

**Figure 2.** (a) Simplified scheme of the elevation profile at the line $\overline{AE}$ in Figure 1c, inset (ii). (b) Diagram of the radar pulse, $I_a$, backscattered from the ground. (c) Diagram of the radar pulses backscattered from the ground, $I_a$, and the makeshift shelter, $I_b$. (d) Red lines: SAR backscattering intensity time series at some representative points of the recently invaded land in Morro Solar. The sudden increment from April 14 denotes the change of the backscatter mechanism from the mechanism shown in 2(b) to that shown in 2(c); Blue lines: SAR backscattering intensity time series at some points of an old informal settlement located close to the recently invaded land.

(Figure 2b). In the presence of an object (Figure 2c), the radar pulses bounces-off the ground towards the object, and then it is reflected from the object to the sensor. Therefore, a larger fraction of the energy is backscattered to the sensor. In the case of the new informal settlements, such objects are indeed the makeshift shelters. We exploit this pattern to map the extent of the occupied area. Figure 2d shows the time series backscattered intensity at some representative points of the recently occupied areas (red lines) and existing built-up areas (blue lines). As expected, a clear increment is observed in the invaded area from April 14. 2021. Therefore, to map the recently occupied areas, a threshold for each pixel coordinate is computed. The threshold is set as the average plus two times the standard deviation of the time series backscattering intensity until April 2, 2021. Then, if the backscatter intensity recorded on April 14, 2021, is greater than the threshold, it is assumed that the area is occupied.

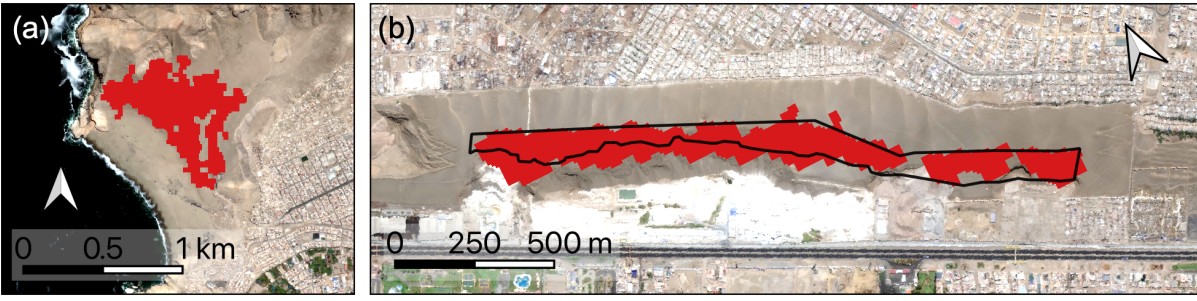

**Figure 3.** Sentinel-1 SAR-based map of the recent informal settlement in Morro Solar (a), and Lomo de Corvina (b). Basemaps: MAXAR imagery

After thresholding, the morphological operators *closing* and *opening* were applied using a kernel size of $3 \times 3$. Then, pixel clusters were identified, and those with sizes lower than 200 pixels were filtered out. Figure 3 depicts the extent of the invaded areas in Morro Solar and Lomo de Corvina. The black polygon shown in Figure 3b denotes the extension of the occupied area estimated from visual inspection of images recorded by an unmanned aerial vehicle (UAV). Our results from SAR images identified 84% of the area identified by visual inspection. Furthermore, only 67% of the area identified from SAR images is contained within the black polygon. We believe that the main reason for the discrepancies between SAR images' results and that from visual inspection lies in the complex geometric distortions in the SAR images.

The results show that a recent squatter settlement could be identified from radar images. Considering the open access to Sentinel-1 imagery, it represents an opportunity to implement a sustainable, low-cost system to monitor informal urban growth over unsafe areas and perform hazard mitigation actions, such as relocation. Prompt measures can prevent significant losses to society. For instance, the mapped settlements are 438,000 m$^2$ and 265,300 m$^2$ in Morro Solar and Lomo de Corvina, respectively. From a visual inspection of the urban settlements nearby the recently occupied areas, we estimated a total of 3051 houses in Morro Solar and 1595 in Lomo de Corvina that would have been exposed to strong ground motion amplification and potential landslides during earthquakes.

## 4    Conclusions

Informal urban growth is a recurrent problem in Peru that has increased in recent years. When the settlement is located in a hazardous area, an assessment of the extent of the occupation can be valuable to perform proper actions. In this study, we analyzed a recent informal settlement in two areas in Lima, the capital of Peru, through SAR images of 10 m pixel resolution. A time-series analysis was performed to identify increments of the backscattering intensity at the areas occupied by the settlements. An increment in the SAR intensity is observed from April 14, 2021, which is consistent with local media information regarding the dates of the invasion. We identified 438,000 m$^2$ occupied in the Morro Solar, and 265,300 m$^2$ at Lomo de Corvina. One limitation is that the area identified at Lomo de Corvina is affected by the geometric distortions in SAR images. A

future extension of this work will implement an automated monitoring system of informal settlements in unsafe areas against natural hazards. The monitoring system, constrained by the acquisition dates of Sentinel-1 images, can identify settlements within few days. However, through a potential integration of other satellite constellations, a near-real time monitoring system can be achieved. It is worth mentioning that this study focused only on the natural hazard aspects of informal urban growth, which might be a narrow view of the problem. We believe, however, this study will be valuable for the authorities that must have a general view on the issue of informal urban growth.

*Author contributions.* LM, FG, CG, and MD conceived and designed the study. All authors contributed in the analysis of the data, discussion of the results, and preparation of the manuscript.

*Competing interests.* The authors declare that they have no conflict of interest.

*Acknowledgements.* This work was partly supported by the Concytec–World Bank research program "Improvement and Extension of the services of the National System of Science, Technology and Technological Innovation" 8682-PE through its executing unit PROCIENCIA [contract 038-2019], the JSPS Kakenhi 21H05001, and the TCPAI Center and the Core Research Cluster of Disaster Science at Tohoku University (a Designated National University).

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
