# Peer review of "Brief communication: Radar images for monitoring informal urban settlements in vulnerable zones in Lima, Peru"

_Natural Hazards and Earth System Sciences, 2021_

## Author Comment (AC1)

**Response to the Comments of Reviewer No. 1**

We sincerely acknowledge the reviewer for her/his time spent reviewing this manuscript. The manuscript has been improved after addressing the reviewer's comments. The following contains our response to each comment. Each response contains a reply and reports the changes in the manuscript. In this document, we refer to the manuscript that was revised by the reviewer as the *original* manuscript, and the manuscript that contains the modifications, based on the reviewer's comments, is referred as the *updated* manuscript.

**Comment:**
"Line 40 in Page 2

"However" is a bit strange from the context of the paragraph. Please correct it."

**Reply:**
As suggested by the reviewer, we have rephrased the sentence.

**Change in manuscript:**
Line 40:

> were delimited with chalk, and makeshift shelters were built with wooden sticks and plastic sheets (See Figures 1e and 1f). It is worth mentioning that the government declared the Morro Solar as an intangible area in 1977 and a historical monument in
>
> 40  1986. The police, in coordination with the local authorities, removed the inhabitants that participated in the invasions in April 14 and April 28 in Morro Solar and Lomo de Corvina, respectively, and the makeshift shelters were removed several days later.

**Comment:**
"Figure 1(c)-(i), -(ii), (d)-(i), -(ii) and Figure 3

Please add north arrows in each figure."

**Reply:**
As suggested by the reviewer, we include the north arrows in the referred figures.

**Change in manuscript:**
Figure 1:

[Figure]

**Figure 1.** Location of the study area. (a) Location of Metropolitan Lima (blue rectangle) within Peru. (b) Location of the study area (blue rectangle) within Metropolitan Lima, which is partly located in the districts of Chorrillos and Villa el Salvador. (c) Seismic microzonation of the study area. The green rectangle denotes the location of inset (i), the Morro Solar. The blue rectangle denotes the location of the inset (ii), the Lomo de Corvina. (d) Color composite of SAR backscattering intensity images. Red band: image recorded on April 14, 2021; Green and blue band: image recorded on December 03, 2020. (e) Photograph of the squatter settlement in Morro Solar recorded by Gestion (2021). (f) Photograph of the squatter settlement in Lomo de Corvina (modified from Inga (2021)).

Figure 3:

[Figure]

**Figure 3.** Sentinel-1 SAR-based map of the recent informal settlement in Morro Solar (a), and Lomo de Corvina (b).

Comment:
"Figure 3 in Page 4

Considering the spatial resolution of Sentinel-1 images as shown in Fig. 1, Pixel size of the extraction results in Fig. 3 seems much larger than the original resolution. Furthermore, there are no misdetection in outer area of the target area. If the author performed further analysis other than thresholding, please describe it."

Reply:
The reviewer is right, performed further analysis to reduce the effect of the speckle noise in SAR images. We apologize for missing such details. The further processing is explained as follow: After the thresholding (Figure R1a), the morphological operators "opening" and "closing" with a kernel-size of 5x5 pixels were applied (https://docs.opencv.org/master/d3/dbe/tutorial_opening_closing_hats.html), see Figure R1b. Then, we identified the pixel clusters, where a cluster is a set of neighbor pixels, and two pixels are neighbor if they share a common side (Figure R1c). finally, clusters consistent of less than 200 pixels were filtered out. Note that a pixel has an area of about 100m2, which is less than the average area of informal buildings in Lima; furthermore, the number of people that participate in such informal occupations is in the order of thousands. Therefore, we consider that a threshold size of 200 pixels to filter out small clusters is a reasonable choice. Note also that the final results (Fig R1d) include additional clusters that are easy to discriminate because some (the three small clusters on the left) are located in the sea and some are located in existing urban areas (the two small clusters on the right).

[Figure]

Figure R1. (a) Binary thresholding. (b) Closing and opening operators with kernel-size 5x5. (c) Clustering. (d) Clusters of less than 200 pixels are filtered.

In the updated manuscript, we decided to apply the morphologic operators (opening and closing) with a kernel size of 3x3 (instead of 5x5) to improve the final resolution:

[Figure]

Figure R2. (a) Binary thresholding. (b) Closing and opening operators with kernel-size 5x5. (c) Clustering. (d) Clusters of less than 200 pixels are filtered.

Change in manuscript:

Line 94:

the backscatter intensity recorded on April 14, 2021, is greater than the threshold, it is assumed that the area is occupied. After
95    thresholding, the morphological operators *closing* and *opening* were applied using a kernel size of $3 \times 3$. Then, pixel-clusters
were identified and those with size lower than 200 pixels were filtered out. Figure 3 depicts the extent of the invaded areas in
Morro Solar and Lomo de Corvina. The black polygon shown in Figure 3b denotes the extension of the occupied area estimated
from visual inspection of images recorded by an unmanned aerial vehicle (UAV). Our results from SAR images identified 84%
of the area identified by visual inspection. Furthermore, only 67% of the area identified from SAR images is contained within
100   the black polygon. We believe that the main reason of the discrepancies between the information from SAR images and visual
inspection lies in the complex geometric distortions in the SAR images.

Figure 3:

[Figure]

**Figure 3.** Sentinel-1 SAR-based map of the recent informal settlement in Morro Solar (a), and Lomo de Corvina (b).

Comment:

"Line 83-84 in Page 5

The author described "the occupied area in Lomo de Corvina is underestimated" without any evidence. Underestimation must be judged from comparison with other investigation such as field survey or previous studies. Please show some evidence for the underestimation."

Reply:

The reviewer is right, the referred statement was based on the typical geometric distortions of SAR images in areas with significant slope as we expected to be the case of the Lomo de Corvina. However, recent evidence published in the internet proved that it is not the case.

The geometric distortions that occur in Lomo de Corvina area is referred as foreshortening/layover. Such distortions are well-known effects in SAR images (See, for instance, section *2.1.4 Geometric Properties of SAR data* of the book *The SAR Handbook:* https://servirglobal.net/Global/Articles/Article/2674/sar-handbook-comprehensive-methodologies-for-forest-monitoring-and-biomass-estimation). Figure R3 shows Lomo de Corvina recorded from both an optical sensor and a microwave sensor. Grid lines are drawn for better visualization of the geometric distortions. The bare land, which was recently occupied, seems smaller in the SAR image that that from the optical image. Because the Lomo de Corvina area was shrunk in the SAR images, we stated our results might be underestimated.

[Figure]

Figure R3. Lomo de Corvina recorded from an optical sensor (left), and from a microwave sensor (right).

However, videos recorded from a UAV at the occupied areas in Lomo de Corvina were recently published in the internet:

- https://www.dreamstime.com/lima-peru-zone-known-as-lomo-de-corvina-people-illegal-invasion-land-poor-people-illegal-land-dealer-lima-lima-peru-april-video217521408
- https://www.dreamstime.com/lima-peru-zone-known-as-lomo-de-corvina-people-illegal-invasion-land-poor-people-illegal-land-dealer-lima-lima-peru-april-video217520606
- https://www.dreamstime.com/lima-peru-zone-known-as-lomo-de-corvina-people-illegal-invasion-land-poor-people-illegal-land-dealer-lima-lima-peru-april-video217520563
- https://www.dreamstime.com/lima-peru-zone-known-as-lomo-de-corvina-people-illegal-invasion-land-poor-people-illegal-land-dealer-lima-lima-peru-april-video217520465
- https://www.dreamstime.com/lima-peru-zone-known-as-lomo-de-corvina-people-illegal-invasion-land-poor-people-illegal-land-dealer-lima-lima-peru-april-video217520257
- https://www.dreamstime.com/lima-peru-zone-known-as-lomo-de-corvina-people-illegal-invasion-land-poor-people-illegal-land-dealer-lima-lima-peru-april-video217518049
- https://www.dreamstime.com/lima-peru-zone-known-as-lomo-de-corvina-people-illegal-invasion-land-poor-people-illegal-land-dealer-lima-lima-peru-april-video217517546
- https://www.dreamstime.com/lima-peru-zone-known-as-lomo-de-corvina-people-illegal-invasion-land-poor-people-illegal-land-dealer-lima-lima-peru-april-video217516965
- https://www.dreamstime.com/lima-peru-zone-known-as-lomo-de-corvina-people-illegal-invasion-land-poor-people-illegal-land-dealer-lima-lima-peru-april-video217516619
- https://www.dreamstime.com/lima-peru-zone-known-as-lomo-de-corvina-people-illegal-invasion-land-poor-people-illegal-land-dealer-lima-lima-peru-april-video217515700
- https://www.dreamstime.com/lima-peru-april-th-aerial-media-over-pan-american-highway-one-most-important-america-crossing-south-to-north-video217513019
- https://www.dreamstime.com/lima-peru-zone-known-as-lomo-de-corvina-people-illegal-invasion-land-poor-people-illegal-land-dealer-lima-lima-peru-april-video217512728

Figure R4 shows an image of the northwest side of the occupied area, which is limited by a white wall. A simplified idealization of a cross-section in Lomo de Corvina hill is depicted in Figure R5. The location of the cross-section is depicted in Figure R4 as the segment $\overline{AE}$. The cross-section is simplified as four segments $\overline{AB}$, $\overline{BC}$, $\overline{CD}$, and $\overline{DE}$. We assume that the range direction is parallel to the vertical plane that contains the cross-section to illustrate the geometric distortions in the SAR image. The *slant range* denotes the direction from which the microwave energy travels from the satellite to the ground. The microwave images are presented in *ground-range* format. Note that because the points A and C share about the same distance to the satellite, both segments $\overline{AB}$ and $\overline{CB}$ occupy the same geographic position in the ground-range (see $\overline{ab}$ and $\overline{cb}$). In the ground-range the cross-section is located along the segment $\overline{AE'}$ in the optical image; on the other hand, the cross-section is located along the segment $\overline{ae}$ in the microwave image. It is observed that the length of $\overline{AE'}$ is larger than the length of $\overline{ae}$, which is the reason we stated our results might be underestimated. However, the reviewer is right mentioning that our assumption is not conclusive. Note that the length of $\overline{DE'}$, the segment that contains the occupied area, is slightly larger than the length of $\overline{de}$.

[Figure]

Figure R4. Occupied area in Lomo de Corvina (Source: internet)

[Figure]

Figure R5. Simplified scheme of the cross-section of the Lomo de Corvina hill. It is assumed that the cross section and the range direction lie on the same vertical plane. Due to the oblique observation of SAR images, surface slopes lead to geometric distortions.

Regarding a comparison of our results and field surveys, by georeferencing some images extracted from the videos, and with the aid of a high-resolution optical images, we manually draw the extent of the occupied areas at the Lomo de Corvina (See Figure R6a). Then we shifted the polygon to fit the occupied area in the SAR image, we use as reference the boundary between the existing urban area and the recently occupied area (See Figure R6b).

[Figure]

Figure R6. Extent of the occupied area in Lomo de Corvina

Figure R6c shows the estimated occupied areas from SAR imagery and that from visual inspection (VI). 84% of VI-based area were identified in the SAR-based results; on the other hand, 67% of the SAR-based results belong to the VI-based area.

Change in manuscript:
Line 71:

70  color composite of backscattered intensity recorded on different dates. The red band denotes an image recorded on December 03, 2020, and the green and blue band denote the image recorded on April 14, 2021. Note from inset (ii) in Figure 1c and 1d that the Lomo de Corvina area in the SAR image looks smaller than that from the optical image. Such geometric distortions are because of the oblique observation geometry of SAR images. Figure 2a depicts a scheme of the elevation profile corresponding to the line $\overline{AE}$ depicted in the inset (ii) of Figure 1. The profile is simplified by four segments, $\overline{AB}$, $\overline{BC}$, $\overline{CD}$, and $\overline{DE}$. In

75  order to illustrate the geometric distortions in the SAR images, it is assumed that the range direction is parallel to the vertical plane that contains the profile. The slant range denotes the direction from which the microwave energy travels from the satellite to the ground. The microwave images are presented in ground-range format. Note that because the points A and C share about the same distance to the satellite, both segments $\overline{AB}$ and $\overline{CB}$ occupy the same geographic position in the ground-range ($\overline{ab}$ and $\overline{cb}$) of the SAR image. In the optical image, the profile scheme is located over the line $\overline{AE}$ (Figure 1c); on the other hand, the

80  profile is located along the segment $\overline{ae}$ in the SAR image, which is significantly smaller. However, the segment that contains the informal settlement in the optical image, $\overline{DE}$, is only slightly larger than the length of $\overline{de}$. Note also there is a shift between $\overline{DE}$ and $\overline{de}$.

Figure 2:

[Figure]

**Figure 2.** (a) Simplified scheme of the elevation profile at the line $\overline{AE}$ in Figure 1c, inset (ii). (b) Diagram of the radar pulse, $I_a$, backscattered from the ground. (c) Diagram of the radar pulses backscattered from the ground, $I_a$, and the makeshift shelter, $I_b$. (d) Red lines: SAR backscattering intensity time series at some representative points of the recently invaded land in Morro Solar. The sudden increment from April 14 denotes the change of the backscatter mechanism from the mechanism shown in 2(b) to that shown in 2(c); Blue lines: SAR backscattering intensity time series at some points of an old informal settlement located close to the recently invaded land.

Line 97:

95    thresholding, the morphological operators *closing* and *opening* were applied using a kernel size of $3 \times 3$. Then, pixel-clusters were identified and those with size lower than 200 pixels were filtered out. Figure 3 depicts the extent of the invaded areas in Morro Solar and Lomo de Corvina. The black polygon shown in Figure 3b denotes the extension of the occupied area estimated from visual inspection of images recorded by an unmanned aerial vehicle (UAV). Our results from SAR images identified 84% of the area identified by visual inspection. Furthermore, only 67% of the area identified from SAR images is contained within

100    the black polygon. We believe that the main reason of the discrepancies between the information from SAR images and visual inspection lies in the complex geometric distortions in the SAR images.

---

## Author Comment (AC2)

**Response to the Comments of Reviewer No. 2**

We sincerely acknowledge the reviewer for her/his time spent reviewing this manuscript. We feel the manuscript has been improved after addressing the reviewer's comments. The following contains our response to each comment. Each response contains a reply and reports the changes in the manuscript. In this document, we refer to the manuscript that was revised by the reviewer as the *original* manuscript, and the manuscript that contains the modifications, based on the reviewer's comments, is referred as the *updated* manuscript.

**Comment:**
"This is not cutting-edge work. In fact, it does not present any scientific novelty, nor does it suggest any new methodology. Indeed, it is an application of SAR technology in the field of risk prevention. Therefore, at best, this work can fall within the scope of a "Technical Note" or a "Case Study" (Possibly "Brief communication")."

**Reply:**
The reviewer is right pointing out the manuscript does not present a new method, nor a scientific novelty. We motivation of this manuscript is to disseminate SAR data as a useful tool to monitor informal urban growth, which is critical social issue in Perú, and other developing countries. Therefore, with all due respect, we believe the manuscript does fall in the scope of Brief communication (https://www.natural-hazards-and-earth-system-sciences.net/about/manuscript_types.html).

**Change in manuscript:**
No changes

**Comment:**
"However, even as a "Technical Note" ("Brief communication"), in order to be useful to an interested reader, and possibly less knowledgeable about this technology, the manuscript should provide more insights and sufficiently detailed contents on the ground conditions of the areas imaged and common backscatter intensity values, including the variety of factors on which they depend (types, sizes, shapes and orientations of the scatterers in the target area, soil characteristics, vegetation cover, moisture content of the target area, as well as the incident angles of the radar beam)."

**Reply:**
Following the reviewer's comment, we have included further information of the radar images, such as polarization (VV) and incident angle at the study areas (about 38°). Besides, videos recorded from a UAV at the occupied areas in Lomo de Corvina were recently published in the internet:

- https://www.dreamstime.com/lima-peru-zone-known-as-lomo-de-corvina-people-illegal-invasion-land-poor-people-illegal-land-dealer-lima-lima-peru-april-video217521408

- https://www.dreamstime.com/lima-peru-zone-known-as-lomo-de-corvina-people-illegal-invasion-land-poor-people-illegal-land-dealer-lima-lima-peru-april-video217520606
- https://www.dreamstime.com/lima-peru-zone-known-as-lomo-de-corvina-people-illegal-invasion-land-poor-people-illegal-land-dealer-lima-lima-peru-april-video217520563
- https://www.dreamstime.com/lima-peru-zone-known-as-lomo-de-corvina-people-illegal-invasion-land-poor-people-illegal-land-dealer-lima-lima-peru-april-video217520465
- https://www.dreamstime.com/lima-peru-zone-known-as-lomo-de-corvina-people-illegal-invasion-land-poor-people-illegal-land-dealer-lima-lima-peru-april-video217520257
- https://www.dreamstime.com/lima-peru-zone-known-as-lomo-de-corvina-people-illegal-invasion-land-poor-people-illegal-land-dealer-lima-lima-peru-april-video217518049
- https://www.dreamstime.com/lima-peru-zone-known-as-lomo-de-corvina-people-illegal-invasion-land-poor-people-illegal-land-dealer-lima-lima-peru-april-video217517546
- https://www.dreamstime.com/lima-peru-zone-known-as-lomo-de-corvina-people-illegal-invasion-land-poor-people-illegal-land-dealer-lima-lima-peru-april-video217516965
- https://www.dreamstime.com/lima-peru-zone-known-as-lomo-de-corvina-people-illegal-invasion-land-poor-people-illegal-land-dealer-lima-lima-peru-april-video217516619
- https://www.dreamstime.com/lima-peru-zone-known-as-lomo-de-corvina-people-illegal-invasion-land-poor-people-illegal-land-dealer-lima-lima-peru-april-video217515700
- https://www.dreamstime.com/lima-peru-april-th-aerial-media-over-pan-american-highway-one-most-important-america-crossing-south-to-north-video217513019
- https://www.dreamstime.com/lima-peru-zone-known-as-lomo-de-corvina-people-illegal-invasion-land-poor-people-illegal-land-dealer-lima-lima-peru-april-video217512728

Based on the videos, we prepare a simplified scheme of an elevation profile in Lomo de Corvina and discussed the scattering mechanism in the topography of Lomo de Corvina. We already provided common backscattering intensity values and the backscattering mechanism in the occupied areas (Figure 2c of the original manuscript, and Figure 2d of the updated manuscript).

Regarding vegetation and soil moisture, the vegetation is almost absent in the case study areas, and we are afraid we do not have information about the soil moisture. However, we believe the referred information is out of the scope of the present study.

Change in manuscript:
From line 67 onwards:

65 cloud-covered in most of the days after the invasion. From the beginning of April 2021 until May 5, only one product from the Sentinel-2 constellation recorded on April 20 was cloud-free. On the other hand, synthetic aperture radar (SAR) images are designed to pass through the clouds; thus, we focus on SAR images from the Sentinel-1 constellation. A total of fourteen Sentinel-1 SAR images, recorded from December 3, 2020, to May 8, 2021, were used for the analysis. The images were taken by the VV polarization from the ascending path, and the incident angle at the two study areas are about $38°$. Figure 1d shows a

70 color composite of backscattered intensity recorded on different dates. The red band denotes an image recorded on December 03, 2020, and the green and blue band denote the image recorded on April 14, 2021. Note from inset (ii) in Figure 1c and 1d that the Lomo de Corvina area in the SAR image looks smaller than that from the optical image. Such geometric distortions are because of the oblique observation geometry of SAR images. Figure 2a depicts a scheme of the elevation profile corresponding to the line $\overline{AE}$ depicted in the inset (ii) of Figure 1. The profile is simplified by four segments, $\overline{AB}$, $\overline{BC}$, $\overline{CD}$, and $\overline{DE}$. In

75 order to illustrate the geometric distortions in the SAR images, it is assumed that the range direction is parallel to the vertical plane that contains the profile. The slant range denotes the direction from which the microwave energy travels from the satellite to the ground. The microwave images are presented in ground-range format. Note that because the points A and C share about the same distance to the satellite, both segments $\overline{AB}$ and $\overline{CB}$ occupy the same geographic position in the ground-range ($\overline{ab}$ and $\overline{cb}$) of the SAR image. In the optical image, the profile scheme is located over the line $\overline{AE}$ (Figure 1c); on the other hand, the

80 profile is located along the segment $\overline{ae}$ in the SAR image, which is significantly smaller. However, the segment that contains the informal settlement in the optical image, $\overline{DE}$, is only slightly larger than the length of $\overline{de}$. Note also there is a shift between $\overline{DE}$ and $\overline{de}$.

Figure 2:

[Figure]

**Figure 2.** (a) Simplified scheme of the elevation profile at the line $\overline{AE}$ in Figure 1c, inset (ii). (b) Diagram of the radar pulse, $I_a$, backscattered from the ground. (c) Diagram of the radar pulses backscattered from the ground, $I_a$, and the makeshift shelter, $I_b$. (d) Red lines: SAR backscattering intensity time series at some representative points of the recently invaded land in Morro Solar. The sudden increment from April 14 denotes the change of the backscatter mechanism from the mechanism shown in 2(b) to that shown in 2(c); Blue lines: SAR backscattering intensity time series at some points of an old informal settlement located close to the recently invaded land.

Comment:
"Figure 1 caption reads "Red band: image recorded on April 14, 2021; Green and blue band: image recorded on December 03, 2020"; later on the same figure, lines 65-66 read "The red band denotes an image recorded on December 10, 2020, and the green and blue band denote the image recorded on April 14, 2021""

Reply:
We apologize for the typo. It has been corrected accordingly.

Change in manuscript:
Line 70:

| 70 | color composite of backscattered intensity recorded on different dates. The red band denotes an image recorded on December 03, 2020, and the green and blue band denote the image recorded on April 14, 2021. Note from inset (ii) in Figure 1c and 1d |
|---|---|

Comment:
"Figure 2c) must include units on the vertical axis."

Reply:
Following the reviewer's comment, we have included units on the vertical axis.

Change in manuscript:
Page 5:

[Figure]

**Figure 2.** (a) Simplified scheme of the elevation profile at the line $\overline{AE}$ in Figure 1c, inset (ii). (b) Diagram of the radar pulse, $I_a$, backscattered from the ground. (c) Diagram of the radar pulses backscattered from the ground, $I_a$, and the makeshift shelter, $I_b$. (d) Red lines: SAR backscattering intensity time series at some representative points of the recently invaded land in Morro Solar. The sudden increment from April 14 denotes the change of the backscatter mechanism from the mechanism shown in 2(b) to that shown in 2(c); Blue lines: SAR backscattering intensity time series at some points of an old informal settlement located close to the recently invaded land.

Comment:
"Given the great relevance of the colors and sharpness of figures, improvements are expected (at least) in Figures 1c), 1d) and 3."

Reply:
The figures have been modified according to the reviewer's comment.

Change in manuscript:
Figure 1:

[Figure]

**Figure 1.** Location of the study area. (a) Location of Metropolitan Lima (blue rectangle) within Peru. (b) Location of the study area (blue rectangle) within Metropolitan Lima, which is partly located in the districts of Chorrillos and Villa el Salvador. (c) Seismic microzonation of the study area. The green rectangle denotes the location of inset (i), the Morro Solar. The blue rectangle denotes the location of the inset (ii), the Lomo de Corvina. (d) Color composite of SAR backscattering intensity images. Red band: image recorded on April 14, 2021; Green and blue band: image recorded on December 03, 2020. (e) Photograph of the squatter settlement in Morro Solar recorded by Gestion (2021). (f) Photograph of the squatter settlement in Lomo de Corvina (modified from Inga (2021)).

Figure 3:

[Figure]

**Figure 3.** Sentinel-1 SAR-based map of the recent informal settlement in Morro Solar (a), and Lomo de Corvina (b).

---

## Author Comment (AC3)

**Response to the Comments of Professor Taubenböck**

It is a great honor that professor Taubenböck, an authority on the subject, revised our manuscript. His comments were very educational. We sincerely acknowledge him for the time spent reviewing this manuscript. The following contains our response to each comment. Each response contains a reply and reports the changes in the manuscript. In this document, we refer to the manuscript that was revised by professor Taubenböck as the ***original*** manuscript, and the manuscript that contains the modifications, based on the reviewer's comments, is referred as the ***updated*** manuscript.

**Comment:**

"The first sentence of the introduction could be supported by a citation, e.g. from the United Nations world urbanization prospects."

**Reply:**

We thank the reviewer for sharing the information. Indeed, the data from World Urbanization Prospects supports our statement. In addition to the report World Urbanization Prospects The 2018 revision. The website platform provides personalized graphs:

[Figure]

**Change in manuscript:**

Line 9:

> Urban sprawl in Latin America has been influenced by the people migration from rural areas to the cities, which have produced
>
> high-density urban areas (United Nations, 2019). Informal settlement refers to the organization of people in search of housing
>
> 10    who occupy unused land and perform collective actions to self-resolve their urban and social organization issues (Kapstein

**Comment:**

"Line 9: must it be the "organization of people" or could it be just one person as well"

**Reply:**

We have double checked other definitions, apart from Kapstein and Aranda, and all of them refers to people, not just one person.

**Change in manuscript:**

No changes.

**Comment:**

"Line 11: I would rather write instead of "consisted mainly of informal settlements" something like "was mainly driven by informal urban expansion"."

**Reply:**

Following the reviewer's comment, we have updated the sentence accordingly

**Change in manuscript:**

Line 11:

> 10  who occupy unused land and perform collective actions to self-resolve their urban and social organization issues (Kapstein and Aranda, 2014). The urban growth in the capital city of Peru during the 20th century was mainly driven by informal urban expansion, which was motivated by the government policy of allowing people with low socioeconomic status to occupy unused

**Comment:**

"Line 16: Your statement "Frequently, informal settlements occupy unsafe zones against natural hazards." was even proven in a scientific study recently by Müller et al., titled Misperceptions of predominant slum locations?"

**Reply:**

We acknowledge the reviewer for sharing relevant literature. We have included in the introduction the work of Müller et al. on the use of slope to assess exposure to landslides of morphological slums and formal settlements.

**Change in manuscript:**

Line 17:

> 15  have kept increasing countrywide. According to the Ministry of Housing, Construction and Sanitation (2021), 93% of the urban growth in Peru between 2001 and 2018 consisted of informal settlements. Frequently, informal settlements occupy unsafe zones against natural hazards. For instance, Müller et al. (2020) showed that slum residents are more likely to settle in areas exposed to landslides than formal residents. For instance, during the Niño Costero phenomenon in 2017, 63800 houses were destroyed

**Comment:**

"Line 20 – maybe it would make sense, to mention here additionally a study on urban growth based on EO-data?"

**Reply:**

Following the reviewer's comment, we included the work of Shi et al. (2019): Urbanization in China from the end of 1980s until 2010 – spatial dynamics and patterns of growth using EO-data

> 20  Remote sensing data have been used to extract information from urban and rural areas, such as land cover classification (Geiß et al., 2020), urban growth (Shi et al., 2019), and detection of damaged buildings (Moya et al., 2021). Regarding infor-

**Comment:**

"Line 23 – that studies in the "informal settlements" domain are still relatively scarce has been documented well by the review of Kuffer in 2016. I suggest to cite her study here."

**Reply:**

Once again, we sincerely thank the reviewer for sharing important literature. We included the work of Kuffer et al. 2016 in the updated manuscript.

> consider the temporal effects is reported in Kraff et al. (2020). However, remote sensing studies to identify spontaneous infor-
> mal settlements consisting of makeshift shelters are scarce (Kuffer et al., 2016). The relevance of this task is the geolocation of
> 25  such settlements to perform prompt prevention actions, such a relocation when the settlements are located in vulnerable areas.

**Comment:**

"Figure 1: In my opinion, image 1c (iii) is of high relevance, and needs to be larger. I wonder if figures 1a and 1b are necessary at all?"

**Reply:**

We agree with the reviewer. Figure 1c (iii) has been enlarged and moved to Figure 1e. We have also included another photo of Lomo de Corvina as Figure 1f. Regarding Figures 1a and 1b, it was a request from the editorial team.

[Figure]

**Figure 1.** Location of the study area. (a) Location of Metropolitan Lima (blue rectangle) within Peru. (b) Location of the study area (blue rectangle) within Metropolitan Lima, which is partly located in the districts of Chorrillos and Villa el Salvador. (c) Seismic microzonation of the study area. The green rectangle denotes the location of inset (i), the Morro Solar. The blue rectangle denotes the location of the inset (ii), the Lomo de Corvina. (d) Color composite of SAR backscattering intensity images. Red band: image recorded on April 14, 2021; Green and blue band: image recorded on December 03, 2020. (e) Photograph of the squatter settlement in Morro Solar recorded by Gestion (2021). (f) Photograph of the squatter settlement in Lomo de Corvina (modified from Inga (2021)).

**Comment:**

"Line 57 onwards: "It is worth mentioning that the collapse of the light makeshifts built during the recent invasions may not have represented an effective danger condition to the inhabitants. However, non-engineering masonry houses could have been constructed in the short term if the inhabitants were not removed." – these sentences sound too much as if there has been done something good for the people, which might be true from the perspective of natural hazards, but not from their perspective of the need for shelter. A differentiated classification should be made here."

**Reply:**

The reviewer is absolutely right. The referred sentence, and the whole manuscript, does not address the more relevant issue: the need for residence. We have clarified in the updated version of the manuscript.

> waves along the slope (Gonzales et al., 2019). It is worth mentioning that the collapse of the light makeshifts built during the
> 60 recent invasions may not have represented an effective danger condition to the inhabitants. However, non-engineering masonry houses could have been constructed in the short term if the inhabitants were not removed. Note however that the solely action of removing shelters will exacerbate the need for residence.

**Comment:**
"Line 80 onwards – is there any chance to provide a quantitative accuracy assessment?"

**Reply:**
I am afraid we were not able to perform field surveys on the occupied areas. We were recommended no to go to the occupied areas as there was friction between the local authorities and the people who occupied the areas. Fortunately, videos recorded from a UAV at the occupied areas in Lomo de Corvina were recently published in the internet:

- https://www.dreamstime.com/lima-peru-zone-known-as-lomo-de-corvina-people-illegal-invasion-land-poor-people-illegal-land-dealer-lima-lima-peru-april-video217521408
- https://www.dreamstime.com/lima-peru-zone-known-as-lomo-de-corvina-people-illegal-invasion-land-poor-people-illegal-land-dealer-lima-lima-peru-april-video217520606
- https://www.dreamstime.com/lima-peru-zone-known-as-lomo-de-corvina-people-illegal-invasion-land-poor-people-illegal-land-dealer-lima-lima-peru-april-video217520563
- https://www.dreamstime.com/lima-peru-zone-known-as-lomo-de-corvina-people-illegal-invasion-land-poor-people-illegal-land-dealer-lima-lima-peru-april-video217520465
- https://www.dreamstime.com/lima-peru-zone-known-as-lomo-de-corvina-people-illegal-invasion-land-poor-people-illegal-land-dealer-lima-lima-peru-april-video217520257
- https://www.dreamstime.com/lima-peru-zone-known-as-lomo-de-corvina-people-illegal-invasion-land-poor-people-illegal-land-dealer-lima-lima-peru-april-video217518049
- https://www.dreamstime.com/lima-peru-zone-known-as-lomo-de-corvina-people-illegal-invasion-land-poor-people-illegal-land-dealer-lima-lima-peru-april-video217517546
- https://www.dreamstime.com/lima-peru-zone-known-as-lomo-de-corvina-people-illegal-invasion-land-poor-people-illegal-land-dealer-lima-lima-peru-april-video217516965
- https://www.dreamstime.com/lima-peru-zone-known-as-lomo-de-corvina-people-illegal-invasion-land-poor-people-illegal-land-dealer-lima-lima-peru-april-video217516619
- https://www.dreamstime.com/lima-peru-zone-known-as-lomo-de-corvina-people-illegal-invasion-land-poor-people-illegal-land-dealer-lima-lima-peru-april-video217515700
- https://www.dreamstime.com/lima-peru-april-th-aerial-media-over-pan-american-highway-one-most-important-america-crossing-south-to-north-video217513019
- https://www.dreamstime.com/lima-peru-zone-known-as-lomo-de-corvina-people-illegal-invasion-land-poor-people-illegal-land-dealer-lima-lima-peru-april-video217512728

by georeferencing some images extracted from the videos, and with the aid of a high-resolution optical images, we manually draw the extent of the occupied areas at the Lomo de Corvina (See Figure R1a). Then we shifted the polygon to fit the occupied area in the SAR image, we use as reference the boundary between the existing urban area and the recently occupied area (See Figure R1b).

[Figure]

Figure R6. Extent of the occupied area in Lomo de Corvina

Figure R6c shows the estimated occupied areas from SAR imagery and that from visual inspection (VI). 84% of VI-based area were identified in the SAR-based results; on the other hand, 67% of the SAR-based results belong to the VI-based area.

**Change in manuscript:**
Line 97:

> 95 thresholding, the morphological operators *closing* and *opening* were applied using a kernel size of $3 \times 3$. Then, pixel-clusters
> were identified and those with size lower than 200 pixels were filtered out. Figure 3 depicts the extent of the invaded areas in
> Morro Solar and Lomo de Corvina. The black polygon shown in Figure 3b denotes the extension of the occupied area estimated
> from visual inspection of images recorded by an unmanned aerial vehicle (UAV). Our results from SAR images identified 84%
> of the area identified by visual inspection. Furthermore, only 67% of the area identified from SAR images is contained within
> 100 the black polygon. We believe that the main reason of the discrepancies between the information from SAR images and visual
> inspection lies in the complex geometric distortions in the SAR images.

**Comment:**
"Line 89 "to perform a proper relocation" sounds cynical to me. It should be discussed in a more holistic sense – yes, relocation might be good due to the exposed areas towards natural hazards, but from a social, economic or personal perspective, relocation might be catastrophic, too. So, please discuss this."

**Reply:**
The reviewer is right. This manuscript covers only the natural hazards aspect of informal settlements. We understand that against the problem of informal urban growth our work is limited. Unfortunately, we are not qualified to perform a social/economic analysis of the informal settlements. In fact, our motivation is that our work will be useful to the politicians/decision-makers that have to consider the full aspects of the problem.

Line 120:

| | |
|---|---|
| | within few days. However, through a potential integration of other satellite constellations, a near-real time monitoring system |
| 120 | can be achieved. It is worth mentioning that this study focused only on the natural hazard aspects of informal urban growth, |
| | which might be a narrow view of the problem. We believe, however, this study will be valuable for the authorities that must |
| | have a general view on the issue of informal urban growth. |

---

## Author Response (AR2)

**RESPONSE TO "REMARKS FROM THE PRECEDING REVIEW FILE VALIDATION"**

**Comment:**

"I noticed that your figure 1 contains maps and aerials. For the next revision, I kindly ask you to clarify whether you have created the maps or were they created by a map provider? If the maps were not created by you, please provide in your revised file that the copyright is denoted in the figure itself. If this is not possible, please provide it in the caption. Please see the section "Manuscript composition" in our manuscript preparation guidelines: https://publications.copernicus.org/for_authors/manuscript_preparation.html"

**Response:**

In Figure 1, we used Open Street Map as base maps. Also, we used high resolution optical images from MAXAR in the insets (i) and (ii). In Figure 3 we used MAXAR imagery. As suggested, we included the credits in the updated version of the manuscript.

**Changes in manuscript:**

Figure 1:

[Figure]

**Figure 1.** Location of the study area. (a) Location of Metropolitan Lima (blue rectangle) within Peru. (b) Location of the study area (blue rectangle) within Metropolitan Lima, which is partly located in the districts of Chorrillos and Villa el Salvador. (c) Seismic microzonation of the study area. The green rectangle denotes the location of inset (i), the Morro Solar. The blue rectangle denotes the location of the inset (ii), the Lomo de Corvina. Basemap of insets: MAXAR imagery. (d) Color composite of SAR backscattering intensity images. Red band: image recorded on April 14, 2021; Green and blue band: image recorded on December 03, 2020. (e) Photograph of the squatter settlement in Morro Solar recorded by Gestion (2021). (f) Photograph of the squatter settlement in Lomo de Corvina (modified from Inga (2021)).

Figure 3:

[Figure]

**Figure 3.** Sentinel-1 SAR-based map of the recent informal settlement in Morro Solar (a), and Lomo de Corvina (b). Basemaps: MAXAR imagery